# INTRA-PROMPT PARALLEL DECODING FOR COMMON-CONTEXT QUESTION ANSWERING

## ABSTRACT

In common-context question answering (CCQA) tasks, multiple questions share a common context to base their answers from. However, Large Language Models (LLMs) typically generate each answer using an independent prompt. While existing batching and caching techniques help improve parallelism and reduce repeated computations, the separation of questions across prompts limits the achievable speedup, as modern GPUs are underutilized due to a memory bottleneck during attention. We present Intra-Prompt Parallel Decoding (IPPD), a novel inference method that answers multiple common-context questions in parallel within a single prompt. IPPD directly addresses the bottleneck by efficiently sharing both memory and computation during the attention process, as the next token for every question is decoded in a single inference step. IPPD uses virtual position IDs and attention mask manipulation to generate the same output as standard prompting without requiring fine-tuning or any changes to the LLM architecture. Since all parallelism occurs within a prompt, IPPD is fully compatible with batched inference, even when each prompt features a different context. Our experiments show that IPPD delivers up to 7X the effective throughput as standard decoding without quality degradation, and outperforms state-of-art inference acceleration methods on real-world datasets.

## 1 INTRODUCTION

Text generation has emerged as a core capability of LLMs, powering downstream NLP applications such as dialogue systems (Yi et al., 2025), summarization (Zhang et al., 2025), and question answering (QA) (Yue, 2025). Within the broad landscape of text generation, contextual QA stands out as a representative benchmark task: given a document and a question, a model must not only locate relevant pieces of evidence within the context, but also integrate and reason across them to answer the question. A large subset of QA tasks referred to as *reading comprehension* follows this setting, where multiple questions are asked about the same passage (Lai et al., 2017; Yang et al., 2018; Rajpurkar et al., 2018; Kočiský et al., 2018). We define this type of task for which many questions share the same underlying context (such as a document, narrative, or report) as **common-context QA (CCQA)**. Although LLMs excel at generating free-form answers for a wide range of applications, the computational cost of generative LLMs limits their adoption in large-scale real-world scenarios. This challenge is particularly critical in CCQA: despite heavy overlap in context across questions, autoregressive decoding repeatedly computes the same prompt and passage for each question, wasting GPU resources with redundant computations and amplifying inefficiencies.

Motivated by cost savings, a significant body of literature has focused on improving the efficiency of LLM inference for text generation as well as CCQA. A major performance bottleneck arises from the challenge of fully utilizing highly-parallel GPU architectures, due to the high memory requirements and sequential nature of autoregressive decoding. State-of-the-art serving platforms, such as vLLM (Kwon et al., 2023), implement a variety of advanced methods to address this: Continuous batching dynamically groups incoming requests to maximize GPU utilization, PagedAttention (Kwon et al., 2023) shares the memory-intensive key-value (KV) cache across requests, and Cascade inference (Ye et al., 2024b; Juravsky et al., 2024) shares the compute-expensive attention score calculation of shared prefixes across requests. While these innovations have substantially accelerated inference, the computational cost of processing large, common contexts for millions of queries

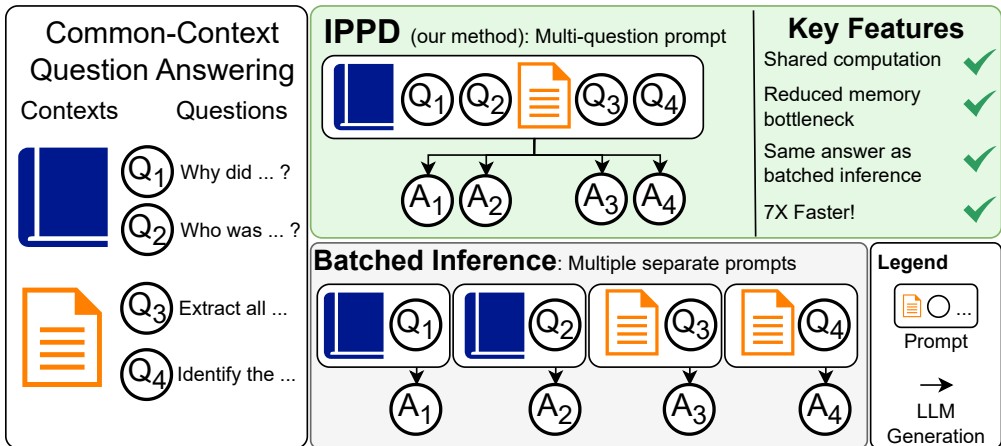

Figure 1: QA tasks frequently include several questions requiring the LLM to reference the same contextual information. Traditional batched inference separates each question into a separate prompt, and processes prompts independently to generate the answer. Our method, IPPD, combines multiple questions and contexts within a single prompt, and jointly decodes multiple answers in parallel. This approach eliminates repeated computations and reduces the memory bottleneck of GPUs during attention to generate the same answers in a fraction of the time.

remains a key challenge, particularly in offline scenarios requiring high throughput, and when not all batched requests share the same context.

To address the aforementioned challenges, we introduce Intra-Prompt Parallel Decoding (IPPD), a novel inference method for accelerating CCQA with intra-prompt parallelism. Figure 1 illustrates the core working principle of IPPD: multiple questions are combined into a single prompt, and answers are decoded in parallel within it. By decoding tokens out of order and adjusting the attention mask, IPPD increases throughput while not affecting the final output. The increased efficiency of intra-prompt parallelism stems from dramatically reducing the number of required memory accesses during attention, which is known to be heavily memory bottlenecked (Recasens et al., 2025). Importantly, IPPD maintains compatibility with orthogonal inference acceleration methods, and is up to 7 times more computationally efficient than standard batched inference on realistic CCQA tasks.

## 2 RELATED WORK

**Token parallelism** Recent efforts to accelerate LLM inference have explored decoding multiple tokens in parallel within a prompt. A common approach is to decode consecutive tokens in parallel. Speculative decoding (Leviathan et al., 2023) employs a smaller draft model to propose a sequence of tokens to be verified in parallel by a larger verifier LLM. The propose-then-verify paradigm is also utilized in Cai et al. (2024) and Fu et al. (2024). A key limitation of these approaches is that the length of the drafted token sequence is directly tied to the level of parallelism, and longer drafts are more prone to quality degradation and subsequent rejection by the verifier model. In contrast, our method parallelizes non-consecutive tokens known to be independent, which eliminates the need for a verification step. Other work has explored parallelizing non-sequential tokens. Ning et al. (2024) and Liu et al. (2024) break down complex tasks into independent sub-tasks that can be processed in parallel. While this can reduce single-task latency, it requires creating and processing multiple new prompts. We show in Section 4.3 that intra-prompt parallelism results in denser computations during attention that more effectively utilize modern GPU hardware compared to batching separate prompts, resulting in higher throughput in offline settings.

**Batch parallelism** Another relevant research direction is accelerating LLM inference by sharing memory and computation across batched prompts. Prefix caching (Kwon et al., 2023; Ye et al., 2024a; Pan et al., 2025) and PagedAttention (Kwon et al., 2023) are common methods that create a shared Key-Value (KV) cache to reduce memory usage and avoid recomputing KV matrices for

common prefixes in batched prompts. While these techniques reduce the required floating point operations (FLOPs) during attention, they are limited by the large number of memory accesses required to independently compute attention scores for each prompt, resulting in GPU underutilization. We elaborate on this phenomenon in Section 4.3. One proposed solution is to separate the attention score calculation of shared prefixes from unique suffixes, in a process known as Cascade inference (Ye et al., 2024b; Juravsky et al., 2024). The prefix attention scores are computed only once, and a merge operator combines them with the unique suffixes scores. However, this approach requires numerous small multiplication operations and suffers from the overhead of the merge operation. In contrast, our method combines the computation of all attention scores into a few larger multiplication operations without any overhead.

## 3   PROBLEM STATEMENT: COMMON-CONTEXT QA TASK

We begin by denoting the inputs to the LLM in a QA setting. All LLM inputs are comprised of the following components: a *common instruction* or prompt prefix $p$, a *context* $c$ (for example a document or passage), and a *question* $x$. Usually, these three logical parts form a triplet $(p, c, x)$ which, after tokenization, becomes a contiguous subsequence presented to the model. In realistic workflows, we can have many such triplets, and several triplets may share the same context. To capture this, let $\{c_j\}_{j=1}^J$ denote contexts and for each context $c_j$ let $\{x_{j,k}\}_{k=1}^{M_j}$ denote the associated questions. The set of logical triplets is then $\mathcal{T} = \{(p, c_j, x_{j,k}) \mid j = 1, \ldots, J, \ k = 1, \ldots, M_j\}$. We assume this set to be fully available offline at inference time.

A traditional LLM loop to process the triplets $\mathcal{T}$ would be to issue $|\mathcal{T}|$ independent prompts, each requiring a separate LLM forward pass $f(\cdot)$. During autoregressive answer generation, each answer token is generated and fed back into the input sequence and passed through forward pass $f$. As a result, the $t$-th token of the answer is obtained using $\hat{y}_{j,k,t} = f(p, c_j, x_{j,k}, y_{j,k,<t})$, where $y_{j,k,<t}$ represents the first $t - 1$ answer tokens generated for question index $k$ of context $c_j$. At the end of the process, we obtain the answer set $\mathcal{A} = \{\hat{y}_{j,k} \mid j = 1, \ldots, J, \ k = 1, \ldots, M_j\}$.

Traditional inference methods employ batching to process multiple prompts concurrently. Although parts of the computation, such as the MLP layers, can be effectively parallelized across prompts, computations in the attention layers are performed independently for each prompt. This introduces a bottleneck for batched inference, particularly when batch elements share a common context $c_j$. We elaborate on this phenomenon in section 4. To overcome this limitation, we wish to develop an inference method that can efficiently share computation in the attention layers, while maintaining batched inference compatibility and not requiring any training or fine-tuning. To achieve this, we aim to parallelize the generation of $\hat{y}_{j,k}$ across $j, k$ *within* a prompt. It is important to note that each token *within an answer* $\hat{y}_{j,k,t}$ will still be generated autoregressively, as we are parallelizing across the *question set*. The core methodological challenges are ensuring that the introduced intra-prompt parallelism is computationally efficient and does not affect the quality of the generated answers $\mathcal{A}$.

We evaluate candidate strategies by comparing against batched autoregressive generation along two complementary axes: *throughput* (efficiency) and *answer quality* (fidelity to the ground truth question answers). Throughput measures how many logical triplets are answered per unit time per GPU. Answer quality is measured at the task level using standard metrics (Accuracy, F1, ROUGE-L). Task-level metrics compare generated answers $\hat{y}_{j,k}$ to ground truth answers $y_{j,k}$. See Appendix B for definitions of throughput and metrics.

## 4   METHODOLOGY: INTRA-PROMPT PARALLEL DECODING (IPPD)

We introduce *Intra-Prompt Parallel Decoding (IPPD)*, a strategy that enables multiple context–question triplets to be processed in parallel within a single prompt. Section 4.1 describes how we replace the $|\mathcal{T}|$ independent forward passes of the canonical autoregressive loop with a single, structured forward-pass strategy that (i) stacks contexts and questions into one prompt, (ii) assigns hierarchical position identifiers so positional relationships do not induce cross-question leakage, (iii) builds an attention mask that enforces per-question causal inference, and (iv) decodes tokens in parallel across the question set. Figure 2 provides an illustrative example of the process: The IPPD prompt combines two contexts $c_1$ and $c_2$ with two questions each, $x_{1,1}, x_{1,2}$ and $x_{2,1}, x_{2,2}$

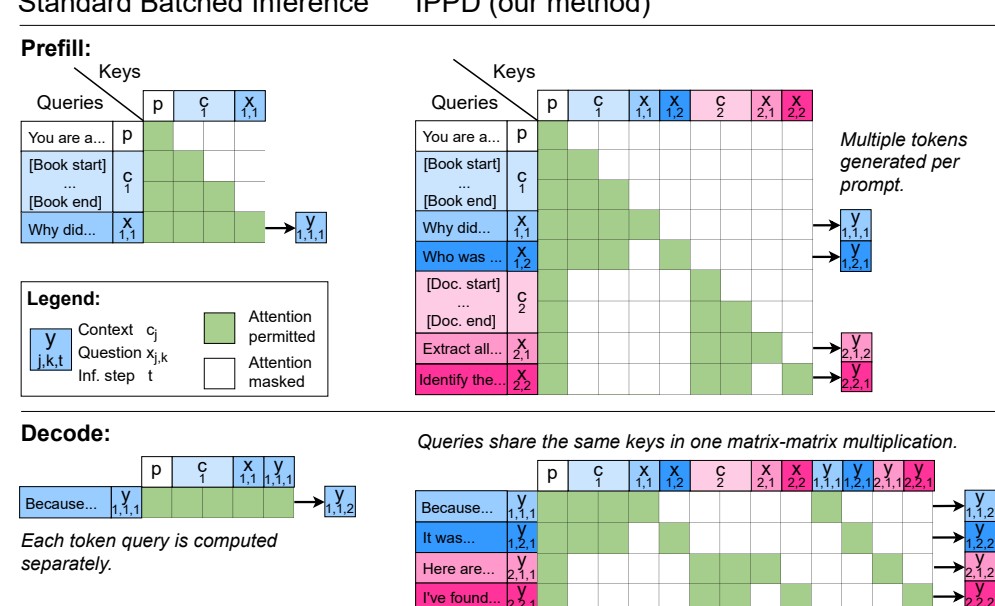

Figure 2: The attention process between the queries (rows) and keys (columns) is depicted with green cells representing unmasked query-key pairs. In standard batched inference, the sequence is composed of a common instruction $p$, shared context $c$, question $x$ and partial answer $y$. Each sub-component is simplified as one or two tokens for illustrative purposes. The arrows indicate which token will be generated using the output of that query row. Our method, IPPD, concatenates multiple contexts and queries into a single stacked prompt. The attention mask ensures that each query can only attend to past tokens from the relevant context and question, identifiable by color and shade respectively. At inference step $t$, IPPD generates the $t$-th token of four answers from a single matrix-matrix multiplication in this example. Batched inference would require four independent attention calculations to achieve the same result.

respectively. During the prefill stage, the attention mask prevents the questions from (a) attending to each other, and (b) attending to the non-relevant context. After a full forward pass, we can extract four answer tokens, corresponding to the first token of each question. These tokens are passed in parallel during the decode stage with appropriate masking to maintain high parallelism throughout generation. Our goal is to ensure that each generated token's conditional distribution under this multi-query, per-step parallel scheme matches the distribution produced by independent autoregressive decoding on the corresponding triplet $(p, c_j, x_{j,k})$, which we demonstrate in Section 4.2. Section 4.3 describes the computational efficiency of IPPD, arising from increasing GPU utilization during attention.

### 4.1 INPUT PRE-PROCESSING

We construct a stacked token sequence $s$ containing the global instruction $p$, $\ell \leq J$ contexts, and the associated questions. We append the answers *after* this header. Concretely, the header region of the prompt is

$$\text{header} = \begin{bmatrix} p, \ c_1, \ x_{1,1}, \ x_{1,2}, \ \ldots, \ c_2, \ x_{2,1}, \ \ldots, c_\ell, \ x_{\ell,1}, \ \ldots, \ x_{\ell,M_\ell} \end{bmatrix}, \tag{1}$$

and the full token sequence $s$ at any point in decoding equals the header followed by the concatenation of all answer tokens produced so far. We do not reserve fixed answer slots inside the header; instead answers are accumulated at the end of the sequence as they are generated.

To make masking causal, we compute a small metadata tuple for each token index $i$ in $s$: a virtual position ID $pos^{\text{virt}}(i)$, a context identifier $context(i) \in \{0, 1, \ldots, J\}$ (with $context(i) = 0$ indicating tokens in $p$), and a question identifier $question(i) \in \{0, 1, \ldots M\}$ (with $question(i) = 0$ for tokens that are global context).

We distinguish between a *virtual position* that encodes the per-triplet autoregressive ordering and the *absolute position* that reflects the token's index in the concatenated sequence $s$. The virtual position of the $t$-th token of the answer for question $(j, k)$ is defined as the position that token would occupy if the triplet $(p, c_j, x_{j,k})$ were decoded in isolation:

$$pos^{\mathrm{virt}}\big(y_t^{j,k}\big) \; = \; |p| \; + \; |c_n| \; + \; |x_{j,k}| \; + \; (t-1), \tag{2}$$

where $|\cdot|$ denotes the token length of a sequence. For header tokens inside $p$, $c_n$, or $x_{j,k}$, their virtual positions $pos^{\mathrm{virt}}(\cdot)$ cover these ranges:

$$pos^{\mathrm{virt}}(p) = [0 \,, \, |p|), \tag{3}$$

$$pos^{\mathrm{virt}}(c_n) = [|p| \,, \, |p| + |c_n|), \tag{4}$$

$$pos^{\mathrm{virt}}(x_{j,k}) = [|p| + |c_n| \,, \, |p| + |c_n| + |x_{j,k}|). \tag{5}$$

For answer tokens, $pos^{\mathrm{virt}}(\cdot)$ gives the local offset inside their parent triplet. Appendix A.1 provides a derivation for the absolute position IDs.

We define the masking logical expressions to use virtual positions for the causality constraint. Let $q, k$ be token indices in $s$ and let $context(\cdot)$ and $question(\cdot)$ be as previously defined. The elementary logical expressions become

$$\mathrm{causalMask}(q, k) \; = \; \big(pos^{\mathrm{virt}}(q) \geq pos^{\mathrm{virt}}(k)\big), \tag{6}$$

$$\mathrm{contextMask}(q, k) \; = \; \big(context(q) = context(k)\big) \; \vee \; \big(context(k) = 0\big), \tag{7}$$

$$\mathrm{questionMask}(q, k) \; = \; \big(question(q) = question(k)\big) \; \vee \; \big(question(k) = 0\big). \tag{8}$$

The final mask logic is the conjunction

$$\mathrm{finalMask}(q, k) \; = \; \mathrm{causalMask}(q, k) \; \wedge \; \mathrm{contextMask}(q, k) \; \wedge \; \mathrm{questionMask}(q, k), \tag{9}$$

and the transformer's attention matrix mask is set by

$$M_{q,k} \; = \; \begin{cases} 0, & \text{if } \mathrm{finalMask}(q, k) \text{ is true,} \\ -\infty, & \text{otherwise.} \end{cases} \tag{10}$$

Note that since the attention block output goes through a softmax layer, negative infinity inputs result in zero activation post-softmax.

Because answers are appended at the end of the sequence and are labeled with their originating question and context, this mask guarantees that any answer token $y_t^{j,k}$ may attend only to the global instruction $p$ (via $context(k) = 0$), to tokens in its parent context $c_j$ (via $context(q) = context(k)$), and to tokens in its own question/answer pair $x_{j,k}$ and previously generated answer tokens (via $question(q) = question(k)$). Tokens belonging to other questions or contexts fail at least one mask and are therefore blocked, preventing cross-question information leakage despite the single forward-pass execution.

## 4.2 PARALLEL DECODING

Building on the hierarchical position and masking scheme described above, we now argue that the resulting forward pass is equivalent to independent autoregressive decoding over each triplet $(p, c_n, x_{j,k})$. The key point is that the attention mechanism for any output token $y_t^{j,k}$ is restricted by construction to exactly the same set of tokens it would see in the autoregressive case: the shared instruction $p$, the corresponding context $c_n$, the question $x_{j,k}$, and its own previously generated tokens $y_{<t}^{j,k}$. Tokens from other questions or contexts are masked and therefore have no influence.

Formally, let $q$ be the query vector for token $y_t^{j,k}$, $K$ the matrix of key vectors over all tokens, and $S$ the set of valid keys for the auto-regressive computation. In the autoregressive setting, we have

$$attn_{\mathrm{AR}}(y_t^{j,k}) \; = \; \mathrm{softmax}\Big(\frac{qK^\top}{\sqrt{d_k}} + M_{\mathrm{CAUSAL}}\Big)_S V, \tag{11}$$

where the softmax is restricted to indices in $S$ using a causal mask $M_{\text{CAUSAL}}$. Under IPPD, the same attention score is computed over all keys but with a mask $M_{\text{IPPD}}$ applied:

$$attn_{\text{IPPD}}(y_t^{j,k}) \;=\; \text{softmax}\Big(\frac{qK^\top}{\sqrt{d_k}} + M_{\text{IPPD}}\Big)V, \tag{12}$$

where

$$M_{\text{IPPD}}[k] \;=\; \begin{cases} 0, & k \in S, \\ -\infty, & k \notin S. \end{cases} \tag{13}$$

Since adding $-\infty$ to a logit removes it from the softmax support, the resulting distribution is identical to that of the autoregressive case:

$$attn_{\text{IPPD}}(y_t^{j,k}) \;=\; attn_{\text{AR}}(y_t^{j,k}). \tag{14}$$

Thus, each token in the stacked prompt follows exactly the same computational path as in the independent autoregressive runs, establishing the equivalence of the two procedures.

Each forward pass at step $t$ generates the $t$-th token of each answer, which is appended to the prompt with the correct virtual position ID for the next pass. We stop appending tokens for completed answers, continuing until all answers are complete. This process is demonstrated in Fig. 2.

### 4.3 COMPUTATIONAL EFFICIENCY

IPPD allows for more efficient computation in the attention layers. We provide a high-level explanation in this section, with a more detailed analysis and comparison to Cascade inference available in Appendix A.2. At each attention layer, the attention computation involves multiplying $q$ and $K^T$, followed by a multiplication with $V$ post-softmax. These large multiplications incur a time cost for each byte of memory accessed to load these matrices, as well as a time cost for each floating point operation (FLOP) required for calculating the output. In autoregressive decoding, attention is almost exclusively bottlenecked by memory accesses, even for small models and large batch sizes (Recasens et al., 2025). This means that modern GPUs are underutilized because the ratio of FLOPs to bytes accessed from memory, known as *arithmetic intensity*, is too low. IPPD alleviates this bottleneck by increasing arithmetic intensity.

Let $K_p, K_{c_j}, V_p, V_{c_j}$ denote the keys and values belonging to the instruction tokens $p$ and the common context tokens $c_j$ respectively. These keys and values can be computed once and cached for all prompts in the batch, in a process known as prefix caching. Nevertheless, for each attention layer, batched autoregressive inference still requires each key and value to be accessed once for each output token, as each prompt requires an independent attention calculation. Since IPPD combines all $\{x_{j,k}\}_{k=1}^{M_j}$ questions into one stacked prompt, only one memory access is required for $K_p, K_{c_j}, V_p, V_{c_j}$ to generate $M_j$ output tokens. As the instruction $p$ and context $c_j$ usually contain the most tokens, reducing the memory accesses for their keys and values by a factor of $M_j$ significantly reduces the memory accesses per token for IPPD. Although the memory accesses are reduced, the larger prompts of IPPD require a larger number of FLOPs per output token. Since attention is memory-bound, this tradeoff is worthwhile and results in higher decoding throughput. Appendix A.2 quantifies this tradeoff, which can be summarized as two key observations:

1. For tasks requiring a **single-token answer**, like multiple-choice QA (the RACE dataset in Table 1), IPPD is most effective with shorter contexts, as arithmetic intensity is lower in this setting.

2. For tasks requiring a **multi-token answer** (the NarrativeQA dataset in Table 1), IPPD is effective for both short and long contexts, as arithmetic intensity is very low regardless of context length.

### 4.4 BATCH INFERENCE COMPATIBILITY

IPPD is fully compatible with batched inference: We can batch $b$ stacked prompts each with $\ell$ contexts. This allows us to trade off having shorter stacked prompts but lower intra-prompt parallelism. When using batched inference with IPPD, special care must be taken to maintain a uniform input length across stacked prompts. During prefill, the initial prompt is left-padded, as is normally done with standard batching. During decoding, each prompt may produce a variable number of output

tokens per inference step. This is because (a) the number of questions or contexts may vary across stacked prompts, and (b) some answers may complete at an earlier step than others, thus reducing the number of outputs in the next step non-uniformly. To address this, we append pad tokens at each decoding step to match the input length of the prompt with the most answers currently being generated. We assign these pad tokens a virtual position ID of infinity (implemented as the maximum long integer value), so that the causal mask in Equation (6) prevents any attention. The virtual position IDs of other tokens also ignore the existence of pad tokens, so the batched output of IPPD is unchanged. We also implement prefix caching within IPPD to reuse the KV cache of the common instruction $p$ across batched prompts, which is useful in few-shot settings.

## 5 EXPERIMENTAL SETUP

**Datasets**   We evaluate IPPD on four common-context question answering datasets (Table 1). We select NarrativeQA (Kočiský et al., 2018) as a challenging dataset where answers are not required to be spans of the input, which is best suited for generative QA. We employ 5-shot prompting following (Liang et al., 2023) to investigate performance on long prefixes. We select SQuAD 2.0 (Rajpurkar et al., 2018) as a purely extractive QA benchmark with shorter contexts. We also evaluate on the multiple-choice benchmarks RACE (Lai et al., 2017) and LongHealth (Adams et al., 2025) to cover a wide range of prefix and answer lengths combinations as well as domains. Table 1 provides the important dataset statistics, while Appendix B.2 contains additional details.

| Dataset | Number of contexts | Avg. Parallelism | Average token length | | | |
|---|---|---|---|---|---|---|
| | | | Instruction | Context | Question | Answer |
| NarrativeQA (5s) | 355 | 29.73 | 2,754.00 | 737.45 | 11.97 | 5.81 |
| SQuAD 2.0 | 1,365 | 8.70 | 47.00 | 177.94 | 12.58 | 3.94 |
| RACE | 1,407 | 3.51 | 40.00 | 344.30 | 39.81 | 1.00 |
| LongHealth | 20 | 20.00 | 73.00 | 11,719.75 | 72.50 | 1.00 |

Table 1: Dataset statistics of the four selected CCQA tasks. The average token length of each subsection of the prompt is shown. (5s) means the instruction $p$ includes 5 few-shot examples, used for NarrativeQA. Average parallelism is the average number of questions sharing a common context.

**Models**   We use state-of-the-art LLMs across a wide range of model sizes. We select the Qwen3 models: 32B, 8B, 4B-Instruct-2507, and 1.7B (Qwen Team, 2025) using non-thinking mode for computational efficiency. We also include Phi-4 14B (Microsoft Research, 2024) and OLMo-2-0325-32B-Instruct (OLMo et al., 2025) to represent other model families. Models larger than 8B parameters are quantized to 4 bits to fit on a single GPU.

**Baselines**   As the current version of IPPD is implemented using HuggingFace Transformers (Wolf et al., 2020), we use Transformers' standard batched inference as our primary baseline. We also compare our method to inference using prefix caching and PagedAttention (Kwon et al., 2023) (PC+PA), using vLLM as a backend. Like IPPD, these methods prevent KV recomputation and efficiently shares KV cache memory across questions with a shared context. Since vLLM is a production-oriented backend, it provides a number of *orthogonal* optimizations not available with Transformers, such as advanced scheduling and custom efficient model code. As our goal is to directly compare IPPD against PC+PA specifically, we adjust the vLLM inference hyperparameters to most closely match our baseline system, with other confounding factors mitigated or removed. Appendix B.3 provides full experimental details, including hyperparameter selection for each method.

**Metrics**   We report throughput as defined in Section 3. NarrativeQA uses ROUGE-L and SQuAD 2.0 uses F1 as a quality metric, while RACE and Longhealth use accuracy. We also measure the exact match rate between the answers generated by IPPD and standard batched inference.

| Model | NarrativeQA (5s) | | SQuAD 2.0 | | RACE | | LongHealth | |
|---|---|---|---|---|---|---|---|---|
| | ROUGE-L | EM% | F1 | EM% | Acc.% | EM% | Acc.% | EM% |
| Qwen3-32B | $0.771_{+0.000}$ | 96.5 | $0.649_{-0.004}$ | 96.9 | $91.1_{+0.0}$ | 99.7 | $87.0_{-0.5}$ | 99.5 |
| OLMo-2-32B | $0.752_{+0.000}$ | 97.6 | $0.714_{+0.000}$ | 98.1 | $90.6_{+0.0}$ | 99.8 | - | - |
| Phi4-14B | $0.702_{-0.004}$ | 91.3 | $0.466_{-0.005}$ | 95.8 | $88.0_{+0.0}$ | 99.9 | $86.5_{+0.5}$ | 99.3 |
| Qwen3-8B | $0.727_{+0.000}$ | 97.5 | $0.598_{-0.001}$ | 97.5 | $88.3_{-0.1}$ | 99.6 | $83.3_{+0.0}$ | 100 |
| Qwen3-4B-Instr. | $0.735_{+0.000}$ | 95.7 | $0.659_{+0.002}$ | 98.1 | $87.4_{+0.0}$ | 99.7 | $83.0_{+0.5}$ | 99.5 |
| Qwen3-1.7B | $0.679_{+0.000}$ | 95.7 | $0.590_{+0.002}$ | 95.5 | $76.7_{-0.1}$ | 99.1 | $69.0_{-0.3}$ | 99.3 |

Table 2: Quality metrics for standard batched inference, with the delta to IPPD outputs shown in the subscript. EM (Exact Match)% is defined as the percentage of batched inference and IPPD answers which match each other exactly. Despite IPPD providing the same computational path for each token, numerical instabilities result in a near but not 100% EM rate.

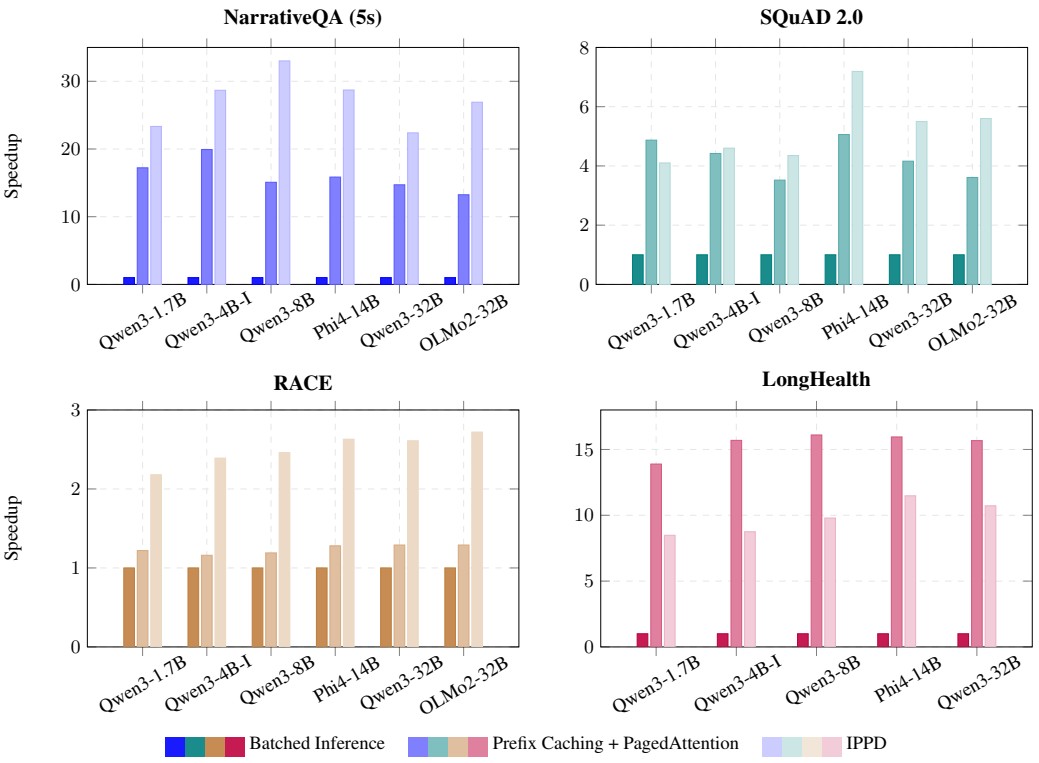

Figure 3: Throughput of each method on a reference g6e.8xlarge EC2 instance, measured in questions answered per second. Normalized throughput is shown relative to the standard batched inference baseline. IPPD outperforms prefix caching + PagedAttention for both short answer QA tasks (top), and for one of the two multiple-choice tasks (bottom). Darker colored shades represent batched inference (baseline), medium shades represent prefix caching + PagedAttention, and lighter shades represent IPPD.

# 6 RESULTS

Table 2 presents the performance comparison between batched inference and IPPD across six models (three model families) ranging from 1.7B to 32B parameters on four CCQA datasets. Results show that IPPD maintains nearly identical performance to batched inference across all benchmarks. Performance differences are less than or equal to 1%, using ROUGE-L on NarrativeQA, accuracy on LongHealth and RACE, and F1 on SQuAD 2.0. This consistency is further validated by high exact match rates between batched inference and IPPD outputs, ranging from 91.3% to 100% across all experiments, with 22 out of 23 model-dataset combinations exceeding 95%. The small variation

in outputs can be attributed to numerical instabilities in the inference process. Overall, these results confirm that IPPD does not affect the output of the LLM in any meaningful way.

Figure 3 showcases the throughput of IPPD and prefix caching + PagedAttention (PC+PA) relative to standard batched inference across datasets and models. Absolute throughput numbers are available in Appendix C. For the short answer generation datasets, IPPD consistently outperforms both batched inference and PC+PA. NarrativeQA throughput increases by up to 32X with Qwen3-8B. Part of this speedup can be attributed to prefix caching of the few-shot examples, but IPPD still outperforms PC+PA by 2.2X, which features more advanced cache management due to the vLLM backend. IPPD also performs strongly with SQuAD 2.0, which features a much shorter instruction and average context length than NarrativeQA. IPPD's relative throughput is consistent across models, ranging from 4.1-5.6X for all models except Phi4-14B, where we measure a 7.2X relative throughput. Although IPPD performs well across the entire model range, it generally performs better relative to PC+PA on larger models, as shown by throughput on SQuAD 2.0 being slightly lower with Qwen3-1.7B but convincingly higher for models with 8 billion parameters or more. This trend suggests that IPPD becomes increasingly valuable as models grow larger, and attention computations scale. Overall, these results demonstrate IPPD's versatility for short-answer CCQA with varying context lengths and model sizes.

For our multiple-choice datasets, IPPD consistently outperforms standard batched inference. The RACE datasets contains the lowest average number of questions per contexts, which results in a moderate 2.1-2.7X relative throughput increase for IPPD. However, PC+PA fails to provide a substantial throughput increase, likely due to the short average context length limiting the effects of KV caching and limiting arithmetic intensity. By stacking six contexts per prompts, IPPD provides a 5.8X greater increase in throughput (+171% v.s. +29%) with OLMo2-32B. This result validates our theoretical analysis in Appendix A.2, which states that that attention can still be memory bottlenecked during prefill when contexts are short. In contrast, LongHealth features extremely long context lengths, which increases the arithmetic intensity during prefill. IPPD is not as effective in this setting as PC+PA, as the advantages of efficient KV cache management for long prefixes outweighs the memory access reductions of IPPD when prefill is already arithmetically intense.

## 7 LIMITATIONS

IPPD is implemented on top of HuggingFace Transformers for ease of experimentation and low-level control during inference. However, it is not optimized for high-throughput serving applications, like is the case with vLLM. As such, the reported inference speed likely understates IPPD's attainable efficiency. We leave the integration of IPPD with vLLM as a promising opportunity for future work. Since IPPD modifies the attention mask, it is not directly compatible with FlashAttention (Dao, 2023). However, new implementations such as FlexAttention (Dong et al., 2024) now provide similar benefits to FlashAttention with the flexibility of custom attention masks. IPPD is implemented with support for both FlexAttention and standard scaled dot-product attention (SDPA), since the latter performs better in most of our experiments. This may change in the future as support for FlexAttention matures and becomes better optimized.

## 8 CONCLUSION

In this paper, we introduce IPPD, a novel approach that accelerates LLM inference for tasks with shared context, such as CCQA. By manipulating position IDs and attention masks, IPPD can decode the next token for all questions in parallel without requiring any modifications to the LLM architecture. The full compatibility with batched inference further enhances IPPD's practical applicability, allowing it to handle multiple prompts with different contexts simultaneously. IPPD achieves up to 7X speedup without sacrificing model performance. Experimental results demonstrate that IPPD consistently outperforms both batched autoregressive decoding and prefix caching with PagedAttention across several benchmark datasets and model sizes. While our experiments focus on CCQA tasks, the core concept of IPPD can extend to a broad range of shared-context generation tasks, such as recommendation systems, summarization, multi-aspect information extraction, etc. Future work could explore combining IPPD with orthogonal efficiency techniques provided by more advanced systems such as vLLM to further reduce inference costs.

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

APPENDIX

# A  METHOD DETAILS

## A.1  POSITION IDS AND ATTENTION MASK

In the stacked sequence $s$, each token has an *absolute position* determined by its literal index in the concatenated prompt. As answers are appended sequentially, absolute positions grow continuously across all contexts and questions. Formally, the absolute position of the $t$-th token of the $k$-th question under context $j$ is

$$pos\big(y_t^{j,k}\big) \;=\; L_{\mathrm{hdr}} \;+\; \sum_{\substack{(j',k')\in\mathcal{Q} \\ \mathrm{rank}(j',k')<\mathrm{rank}(j,k)}} T_{j',k'} \;+\; (t-1) \;+\; 1, \tag{15}$$

where $L_{\mathrm{hdr}}$ is the length of the instruction plus all context and question headers, $\mathcal{Q} = \big((1,1),(1,2),\ldots,(J,M_J)\big)$ is the fixed ordering of question pairs in the prompt, and $T_{j',k'}$ is the token length of the generated answer for question $(j',k')$. This absolute index reflects the physical layout of the stacked sequence.

In contrast, our method introduces a *virtual position* $pos^{\mathrm{virt}}(\cdot)$, which maps tokens back to positions that are local to their own instruction, context, and question triplet. For a generated token $y_t^{j,k}$, the virtual position is

$$pos^{\mathrm{virt}}\big(y_t^{j,k}\big) \;=\; |p| \;+\; \mathrm{len}(c_j) \;+\; |x_{j,k}| \;+\; (t-1). \tag{16}$$

Header tokens are similarly assigned virtual positions aligned with their logical block boundaries. By mapping absolute indices into logically local virtual positions, we ensure that attention remains causal within each sequence, even if two sequences are physically distant in $s$. In other words, $pos^{\mathrm{virt}}(\cdot)$ preserves the *illusion* that each triplet $(p, c_j, x_{j,k})$ is decoded autoregressively in isolation, while still allowing parallel execution within a single stacked prompt.

## A.2  COMPUTATIONAL EFFICIENCY

We provide a deeper analysis of the FLOP and memory access trade-off of IPPD against prefix caching and PagedAttention (PC+PA). For this section, we assume that IPPD uses one context $c_j$, with $M_j$ questions $\{x_{j,k'}\}_{k'=1}^{M_j}$ stacked in one prompt. We calculate how FLOPs and memory accesses scales with the length of $p$, $c_j$, and $x_{j,k}$, which we write as $|\cdot|$. We consider only one attention head of one attention layer, as the computation pattern for each head is the same. The FLOPs and bytes of memory accessed depend on the sequence length of the input and the hidden dimension of the query, key and value associated to each token. Since the hidden dimension is static, we focus on the sequence-length dependent dimension of $q, K$ and $V$, which we denote as $dim(\cdot)$. Table 3 contains $dim(q)$ and $dim(K,V)$ for both methods during different stages of inference. The memory accesses required for each attention computation are linearly proportional to the dimensions of $q, K$ and $V$, so we write that the bytes of memory accessed are $\mathcal{O}(dim(q) + dim(K,V))$, ignoring scaling constants for simplicity. Similarly, the FLOPs required for multiplication during attention are $\mathcal{O}(dim(q) \times dim(K,V))$. The arithmetic intensity is defined as the ratio of FLOPs to memory accesses, which scales as

$$\mathcal{O}\left(\frac{dim(q) \times dim(K,V)}{dim(q) + dim(K,V)}\right).$$

Low arithmetic intensity signifies that memory accesses are causing a performance bottleneck, which Recasens et al. show is often the case with LLM inference. The following two sections compare the approximated FLOPs and memory accesses of PC+PA and IPPD for both prefill and decode. For a context $c_j$ and an inference step $t$, we consider the total FLOPs and memory accesses for computing attention for one output token of every question, $\{y_t^{j,k'}\}_{k'=1}^{M_j}$.

### A.2.1  PREFILL

The first prefill with PC+PA for the prompt answering $x_{j,k}$ requires $dim(q) = dim(K,V) = |p| + |c_j| + |x_{j,k}|$, as the prefix cache has not yet been computed. In subsequent prefills, we only

| Method | | Prefill | | Decode |
| --- | --- | --- | --- | --- |
| | | First | Subsequent | |
| PC+PA | $dim(q)$ | $\|p\| + \|c_j\| + \|x_{j,k}\|$ | $\|x_{j,k}\|$ | $1$ |
| | $dim(K,V)$ | $\|p\| + \|c_j\| + \|x_{j,k}\|$ | | $\|p\| + \|c_j\| + \|x_{j,k}\|$ $+\|y_{<t}^{j,k}\|$ |
| | Frequency | Once per $x_{j,k}$ | | Once per $y_t^{j,k}$ |
| IPPD | $dim(q)$ | $\|p\| + \|c_j\| + \sum_{k'=1}^{M_j} \|x_{j,k'}\|$ | $\|c_j\| + \sum_{k'=1}^{M_j} \|x_{j,k'}\|$ | $M_j^*$ |
| | $dim(K,V)$ | $\|p\| + \|c_j\| + \sum_{k'=1}^{M_j} \|x_{j,k'}\|$ | | $\|p\| + \|c_j\|$ $+ \sum_{k'=1}^{M_j}(\|x_{j,k'}\| + \|y_{<t}^{j,k'}\|)$ |
| | Frequency | Once per $c_j$ | | Once per set $\{y_{<t}^{j,k'}\}_{k'=1}^{M_j^*}$ |

Table 3: Summary of the matrix dimensions for queries $q$ and keys/values $K$ and $V$ for a singular attention head, using prefix caching + PagedAttention (PC+PA) and IPPD with one context per prompt. $dim(\cdot)$ denotes the dimension of that varies with sequence length, ignoring the hidden dimension. Prefill denotes the first forward pass of each prompt, with the First column representing the calculation of the prefix cache. Subsequent prefills leverage this cache to avoid recomputations. The decode column represents the autoregressive decoding stage. The Frequency row indicates how often attention must be performed using these matrix dimensions. $M_j$ denotes the number of questions in context $c_j$. The $*$ indicates that the value may decreases after multiple inference steps, as some answers complete generation before others. The table shows that IPPD's stacked prompts requires larger dimensions but fewer attention calculations.

require $dim(q) = \|x_{j,k}\|$ since the instruction and context are in the prefix cache. The most significant FLOP cost is during the first prefill, which scales as $\mathcal{O}((\|p\| + \|c_j\| + \|x_{j,k}\|)^2)$. $M_j$ attention operations are required to prefill all questions, each requiring the full $K$ and $V$. Therefore, the memory accesses scale as $\mathcal{O}(M_j \times (\|p\| + \|c_j\| + \|x_{j,k}\|))$. **For IPPD**, the FLOP cost scales as $\mathcal{O}((\|p\| + \|c_j\| + \sum_{k'=1}^{M_j} \|x_{j,k'}\|)^2)$, since the prompt contains all questions. The memory accesses are only $\mathcal{O}(\|p\| + \|c_j\| + \sum_{k'=1}^{M_j} \|x_{j,k'}\|)$ since only one attention computation is performed for all questions. The tradeoff obtained by IPPD is a *reduction* in memory accesses by a factor of $M_j$, in exchange for an *increase* in FLOPs by a factor of

$$\frac{(\|p\| + \|c_j\| + \sum_{k'=1}^{M_j} \|x_{j,k'}\|)^2}{(\|p\| + \|c_j\| + \|x_{j,k}\|)^2}.$$

There are two scenarios where this is advantageous:

1. When $\|p\| + \|c_j\| >> \sum_{k'=1}^{M_j} \|x_{j,k'}\|$, and $M_j >> 1$, the relative increase in FLOPs is minor compared to reduction in memory accesses.

2. Since arithmetic intensity (FLOPs / mem. access ratio) scales as $\mathcal{O}(\|p\| + \|c_j\| + \|x_{j,k}\|)$, a shorter $p, c_j$ and $x_{k,j}$ results in lower arithmetic intensity. This scenario is more memory bottlenecked, so would benefit more from the reduction in memory accesses.

We observe in Table 1 that the RACE dataset matches both of these scenarios: the contexts are short, but the questions are even shorter. Figure 3 shows IPPD significantly outperforming PC+PA in this task. For the LongHealth dataset, the contexts are extremely large ($> 10,000$ tokens on average), which increases arithmetic intensity linearly. This dataset is therefore more likely to be FLOP bottlenecked, and not benefit as much from IPPD. We observe in our results that PC+PA outperforms IPPD on this dataset. Both RACE and LongHealth require only one token per answer, so their performance characteristics are solely determined by the prefill stage.

### A.2.2 DECODE

During decode, PC+PA contains a single query, as tokens are generated autoregressively. However, that query must attend to all prior tokens, which requires a separate memory access to the keys and

values of $p$ and $c_j$, $x_{j,k}$ and partial output $y^{j,k}_{<t}$. Arithmetic intensity is very low as both the FLOPs and memory accesses scale linearly with the sequence length. With IPPD, the first decode step has $M_j$ queries. Similarly to the prefill stage, FLOPs per token are increased by a factor of

$$\frac{|p| + |c_j| + \sum_{k'=1}^{M_j}(|x_{j,k'}| + |y^{j,k'}_{<t}|)}{|p| + |c_j| + |x_{j,k}| + |y^{j,k'}_{<t}|}.$$

Since only one memory access to $p$ and $c_j$ is required to generate $M_j$ tokens, the memory accesses to long contexts are reduced by a factor of $M_j$. In subsequent decoding steps, some of the $M_j$ answers may complete generation, reducing the parallelism in future steps. We write $M_j^*$ in Table 3 to denote this effect.

To conclude, the decode stage is characterized by a much lower arithmetic intensity than the prefill stage, resulting in a severe memory access bottleneck. Unlike prefill, the decode stage benefits from IPPD's memory access reduction even for extremely large $p$ and $c_j$. We see in Figure 3 that IPPD performs better than PC+PA for both datasets involving multi-token answers, which range from very short $p$ and $c_j$ in SQuAD 2.0 to very large $p$ and $c_j$ with 5-shot NarrativeQA.

### A.2.3 COMPARISON TO CASCADE INFERENCE

Cascade inference fulfills a similar objective as IPPD: to reduce the number of memory accesses to the common contexts, thereby increasing arithmetic intensity. The way this is achieved is very different from IPPD: Cascade inference computes separate attention scores for keys part of the common prefix and those unique to a prompt. This permits for queries of different prompts to attend to the common prefix keys in a single operation, thus requiring only one memory access to the common prefix keys. Cascade inference also introduces more FLOPs, because a merge operator is required to combine the prefix attention scores with the unique suffixes for each prompt. For settings where the common prefix $p + c_j$ is much larger than the questions, IPPD's FLOPs increase is minimal, as shown in Appendix A.2.1. In settings with long questions or generations relative to the context, the segmentation of attention by Cascade inference would be preferable. The real-world CCQA datasets used in this work more closely match the composition preferred by IPPD, while general few-shot prompting is likely to contain longer suffixes and generations.

## B ADDITIONAL EXPERIMENTAL DETAILS

### B.1 EVALUATION METRIC

We evaluate candidate decoding strategies by benchmarking them against standard batched inference along two complementary dimensions: *throughput* (efficiency) and *answer quality* (fidelity to ground truth answers).

Throughput captures how many logical triplets can be processed per unit time per GPU, directly reflecting efficiency improvements. Concretely, let $|\mathcal{T}|$ denote the number of answered triplets, $T_{\text{wall}}$ the measured wall-clock latency to produce all answers, and $G$ the number of GPUs used. We define

$$\text{Throughput} = \frac{|\mathcal{T}|}{T_{\text{wall}} \cdot G}. \tag{17}$$

Answer quality is measured at the task level using standard NLP evaluation metrics, including Accuracy, F1, and ROUGE-L. These metrics compare generated answers $\hat{y}_{j,k}$ to ground truth answers $y_{j,k}$ across all triplets. We use standard definitions for these metrics from the literature.

### B.2 DATASET DETAILS

**NarrativeQA (Kočiský et al., 2018)** NarrativeQA is a reading comprehension benchmark containing books and movie scripts. We use the human-written summaries as document contexts. About 30 human-generated questions/answer pairs per document are provided. We report results on the test set using 5-shot prompting following the HELM benchmark (Liang et al., 2023).

**SQuAD 2.0 (Rajpurkar et al., 2018)** is an extractive reading comprehension task based on the original Stanford Question Answering Dataset. It contains human generated questions about Wikipedia articles, where the answer is a span of the input text. SQuAD 2.0 adds an additional set of unanswerable questions to increase the difficulty of the task. We report results on the publicly available development set, prompting the LLM to write "null" for unanswerable questions.

**RACE (Lai et al., 2017)** RACE is an English reading comprehension dataset derived from Chinese middle and high school exams. Multiple-choice questions are provided for each text passage by human English instructors covering a variety of topics. We report results on the test set, prompting the LLM to respond with the answer letter.

**LongHealth (Adams et al., 2025)** LongHealth is a collection of detailed patient cases each containing multiple discharge notes. Although realistic in form, structure and content, the documents are **entirely fictional**, so the dataset contains no real patient information. It was written by experienced physicians. We report results on the entire dataset, using the Task 1 setting from Adams et al. (2025). We do not truncate documents, and modify their provided prompt to instruct the LLM to only respond with the answer letter.

### B.3 INFERENCE DETAILS

**General Details** We run all our experiments on the Amazon EC2 g6e.8xlarge server using one Nvidia L40S 48GB GPU. We use greedy decoding for all generations, and non-thinking mode for all hybrid models. We also set a maximum output token length for each dataset: 40 for NarrativeQA, 30 for SQuAD 2.0 and 1 for RACE and LongHealth. Phi4-14B, Qwen3-32B and OLMo-2-32B are quantized to NF4 using BitsAndBytes. Tokenization time is excluded from time measurements. The time required to calculate the attention mask for IPPD is included.

**Standard batched inference** We use the Hugging Face Transformers backend with the $generate()$ method and FlashAttention. Prefix caching is not enabled as it is not supported for batched inference. We increase the batch size for each model-dataset pair until throughput saturates or VRAM limits are exceeded.

**Prefix caching + PagedAttention** We use the vLLM backend for optimized support of prefix caching and PagedAttention. Some vLLM features, such as the asynchronous V1 Engine, provide an inherent advantage over Transformers, irrespective of the inference acceleration methods used. To make a direct comparison between IPPD and prefix caching + PagedAttention, we adjust vLLM settings to exclude advantages that IPPD could also benefit from were it also implemented on the same backend. We use the synchronous V0 Engine, enforcing eager mode as well as the use of the Transformers model implementation to match the one used by IPPD. We tune vLLM performance by providing access to all available VRAM, and setting the maximum model length to the longest prompt + output length for each dataset, so as to maximize the automatically managed batch size. FlashAttention is enabled.

**IPPD** Our method is implemented with Hugging Face Transformers and Accelerate. The model code for each LLM is unchanged, as all IPPD functionality is achieved through a custom inference loop. We implement our own prefix caching only for the few-shot examples of NarrativeQA. We do not share the KV cache across batched prompts as it is not supported by Transformers. For each model-dataset pair, we select the optimal number of shared contexts per prompt and batch size. In practice, we find that these hyperparameters are largely independent of model size. The optimal number of contexts per prompt is largely determined by the context length, number of questions, and hardware architecture. Table 4 shows the hyperparameters used for each method. We largely keep the same IPPD hyperparameters across model size as far as VRAM limits allow.

| Model | | NarrativeQA (5s) | | SQuAD 2.0 | | RACE | | LongHealth | |
|---|---|---|---|---|---|---|---|---|---|
| | | B | C/P | B | C/P | B | C/P | B | C/P |
| Batched Inf. | Qwen3-32B | 8 | 1 | 30 | 1 | 5 | 1 | 1 | 1 |
| | OLMo-2-32B | 8 | 1 | 30 | 1 | 5 | 1 | 1 | 1 |
| | Phi-4-14B | 8 | 1 | 30 | 1 | 5 | 1 | 1 | 1 |
| | Qwen3-8B | 20 | 1 | 30 | 1 | 5 | 1 | 1 | 1 |
| | Qwen3-4B-Instr. | 20 | 1 | 30 | 1 | 5 | 1 | 1 | 1 |
| | Qwen3-1.7B | 20 | 1 | 30 | 1 | 5 | 1 | 1 | 1 |
| IPPD | Qwen3-32B | 2 | 3 | 5 | 6 | 2 | 2 | 1 | 1 |
| | OLMo-2-32B | 2 | 3 | 5 | 6 | 2 | 2 | 1 | 1 |
| | Phi-4-14B | 3 | 4 | 6 | 6 | 2 | 2 | 1 | 1 |
| | Qwen3-8B | 3 | 4 | 6 | 6 | 2 | 2 | 1 | 1 |
| | Qwen3-4B-Instr. | 3 | 4 | 6 | 6 | 2 | 2 | 1 | 1 |
| | Qwen3-1.7B | 3 | 4 | 6 | 6 | 2 | 2 | 1 | 1 |

Table 4: Hyperparameter selection for batch size B and contexts stacked per prompt C/P for IPPD and standard batched inference. Hyperparameters are mostly constant across model sizes, except when VRAM limits require a lower batch size.

## C  ADDITIONAL EXPERIMENTAL RESULTS

### C.1  THROUGHPUT MEASUREMENTS

Table 5 presents absolute throughput scores, i.e., questions answered per second (QPS), for all benchmarks.

| Model | | NarrativeQA (5s) QPS | LongHealth QPS | SQuAD 2.0 QPS | RACE QPS |
|---|---|---|---|---|---|
| Qwen3-32B | Batched Inference | 0.48 | 0.18 | 2.86 | 4.52 |
| | PC + PA | 7.06 | **2.82** | 11.90 | 5.84 |
| | IPPD | **10.74** | 1.93 | **15.74** | **11.78** |
| OLMo-2-32B | Batched Inference | 0.54 | - | 3.47 | 4.75 |
| | PC + PA | 7.15 | - | 12.54 | 6.15 |
| | IPPD | **14.53** | - | **19.43** | **12.91** |
| Phi4-14B | Batched Inference | 0.91 | 0.44 | 5.36 | 10.84 |
| | PC + PA | 14.42 | **7.02** | 27.11 | 13.90 |
| | IPPD | **26.13** | 5.05 | **38.55** | **28.55** |
| Qwen3-8B | Batched Inference | 1.75 | 0.71 | 14.25 | 21.18 |
| | PC + PA | 26.39 | **11.43** | 50.10 | 25.17 |
| | IPPD | **57.77** | 6.95 | **62.04** | **52.12** |
| Qwen3-4B-Instr. | Batched Inference | 1.90 | 1.02 | 18.01 | 34.72 |
| | PC + PA | 37.84 | **16.00** | 79.68 | 40.11 |
| | IPPD | **54.48** | 8.92 | **82.88** | **82.98** |
| Qwen3-1.7B | Batched Inference | 4.54 | 2.40 | 35.36 | 77.74 |
| | PC + PA | 78.20 | **33.33** | 172.07 | 94.88 |
| | IPPD | **105.91** | 20.34 | 145.02 | **169.79** |

Table 5: Throughput comparison of all models on four selected CCQA datasets. We compare batched inference, prefix caching + PagedAttention (PC + PA), and our proposed IPPD. We use queries per second (QPS) as the measurement. The highest score for each setting is in **bold**.

