# OpenReview forum: "Intra-Prompt Parallel Decoding for Common-Context Question Answering"
_ICLR.cc/2026/Conference — Submitted to ICLR 2026_

### Official Review · Reviewer_PotS · 2025-10-16

**Soundness:** 3
**Presentation:** 2
**Contribution:** 1
**Rating:** 2
**Confidence:** 4

**Summary:**

This paper focuses on accelerating common-context question answering, where a shared context (such as a document) is followed by multiple independent questions. The paper introduces Intra-Prompt Parallel Decoding (IPPD).
IPPD combines multiple questions sharing a common context into a single, structured prompt. By setting position_ids and attention_mask, IPPD decodes all questions in parallel.
The authors modify existing benchmarks, such as NarrativeQA and SQuAD 2.0, to fit the above structure format, thereby accelerating their evaluation. The authors say that IPPD achieves up to a 7x throughput improvement over standard batched inference and outperforms Prefix-Caching+PagedAttention.

**Strengths:**

1. The underlying logic is sound: Users may ask an LLM multiple questions in a single API call.
2. The experimental results are robust: the experiments show that IPPD outperforms the baselines listed in the paper.

**Weaknesses:**

1. Limited Practical Applicability due to Input Formatting Assumptions: The paper does not specify how this structured input is created from natural user queries. For instance, when a user submits a prompt such as "Given this report, please summarize the key findings, list the involved parties, and tell me the final conclusion," there is currently no proposed mechanism to parse it into the three distinct questions required by IPPD. Without a defined parsing or pre-processing step, the method is confined to offline batch processing of already-structured datasets and is not directly usable in interactive or conversational systems where user input is unstructured. Although the authors claim to only focus on offline scenarios in the introduction, the scenarios are quite limited.
2. Insufficient Experimental Comparison and Novelty Concerns: Cascade Inference is a highly relevant baseline that also targets the shared-prefix, multi-question scenario. The paper discusses Cascade Inference in Appendix A.2.3, acknowledging that it has a similar objective of reducing memory access. However, the paper lacks a direct experimental comparison, and its introduction section provides only a vague description of cascade inference. This makes it difficult to assess the true novelty and relative advantages of the proposed IPPD approach.

**Questions:**

1. Suggest adding a subsection to the methodology or discussion that addresses the pre-processing of unstructured user queries.

---

> ### Author Response · Authors · 2025-11-21
>
> Thank you for your thoughtful feedback and for your confidence in the soundness of our method and results. We have addressed your comments below.
> - **Limited practical applicability due to input formatting assumptions**: As you correctly mentioned, IPPD is intended to be used in offline settings where the input structure is already known. Translating an unstructured user query into CCQA format would require a detailed study, which falls outside the scope of this paper. We want to highlight that many production systems already structure their inputs:
>     1. System prompts are already separable from the rest of the input in agentic systems. This shared context can greatly exceed the output length when it contains detailed instructions, templates, few-shot examples and tool calling specifications.
>     2. Document analysis and processing tasks already contain several separated documents and queries. Companies, hospitals and governments have a need for intelligently parsing and analyzing unstructured documents in their modernization efforts. Mandal et al. have aggregated such task into a recent benchmark and leaderboard. Attribute Value Extraction in the shopping or web search domain (Brinkmann et al., 2024) also contain pre-structured inputs. The large number of documents to be processed in these settings requires cost-efficient, high throughput inference well suited for IPPD.
>     3. Reinforcement learning algorithms such as GRPO (Shao et al., 2024) require multiple candidate outputs from a query, an offline scenario where high throughput is preferable. It is not necessary to separate question and context in this setting because IPPD can generate outputs in parallel by treating the entire prompt as shared context.
>
>     IPPD therefore provides immediate value in these existing workflows.
>
> - **Experimental comparisons and novelty concerns**: We agree that the inclusion of an additional baseline in the form of Cascade inference, or the similar FastTree referenced by Reviewer 1iVT, would strengthen the experimental results of our work. We are actively working on establishing this baseline and plan to revise the paper with this addition.
>
> We hope that our response satisfies all of your concerns with this work, and are happy to engage in further discussion.
>
> **References:**
>
> Souvik Mandal, Nayancy Gupta, Ashish Talewar, Paras Ahuja, Prathamesh Juvatkar, and Gourinath Banda. Idpleaderboard: A unified leaderboard for intelligent document processing tasks. 2025. https://idp-leaderboard.org/
>
> Alexander Brinkmann, Roee Shraga, and Christian Bizer. Extractgpt: Exploring the potential of large language models for product attribute value extraction. In *Information Integration and Web Intelligence: 26th International Conference*, pp. 38–52. Springer-Verlag, 2024.
>
> Zhihong Shao, Peiyi Wang, Qihao Zhu, Runxin Xu, Junxiao Song, Xiao Bi, Haowei Zhang, Mingchuan Zhang, Y. K. Li, Y. Wu, and Daya Guo. Deepseekmath: Pushing the limits of mathematical reasoning in open language models, 2024. arXiv preprint: arXiv 2402.03300.

---

### Official Review · Reviewer_1iVT · 2025-10-27

**Soundness:** 1
**Presentation:** 2
**Contribution:** 2
**Rating:** 2
**Confidence:** 5

**Summary:**

The paper proposes the intra-prompt parallel decoding (IPPD) for common-context question answering (CCQA). This method generates multiple answers in parallel by constructing a compound prompt that includes multiple prompts with shared context, by assigning the correct attention mask.

**Strengths:**

- The evaluation includes multiple QA datasets (NarrativeQA, SQuAD 2.0, RACE and LongHealth).

**Weaknesses:**

- The proposed method is well-known in the literature.

**Questions:**

The proposed IPPD method is common practice in the literature.
Please refer to the following papers that have already proposed similar methods:
1. SpecInfer: Accelerating Large Language Model Serving with Tree-based Speculative Inference and Verification
2. DeFT: Decoding with Flash Tree-Attention for Efficient Tree-structured LLM Inference
3. FastTree: Optimizing Attention Kernel and Runtime for Tree-Structured LLM Inference
4. FlashForge: Ultra-Efficient Prefix-Aware Attention for LLM Decoding

The following are some more details about these related works:
The author discuss the differences between the proposed IPPD method and prefix caching in section 2, which is appreciated.
The author claim the major novelty of IPPD compared to prefix caching is that it can generate multiple answers of multiple questions in parallel, where prefix caching need to recompute the attention scores and only reuse the KV cache.
However, many more other works have studied the parallel decoding of requests with shared context, the CCQA task is just one of the many applications of this idea (a two depth tree with one shared root).
Besides, these works also explore more advanced techniques to further optimize the attention computation, which can serve as a stonger baseline to compare with in this work.

(Minor) I also find the inference experiment setting confusing. The author turn off the asynchronous decoding feature of vLLM since the baseline methods do not support it.
However, this feature is indeed esential to speed up the decoding when there are multiple requests with shared context and I can't see the reason why the baseline methods cannot support it.

---

> ### Author Response · Authors · 2025-11-21
>
> Thank you for engaging with our work and for your thoughtful feedback. We appreciate the inclusion of several references from related work, which we will include in a revised literature review for better completeness. We are aware that the concept of tree-based attention has been studied in prior work, as we discussed in Section A.2.3. The novelty of IPPD is **providing an efficient method for stacking questions with common context into a single prompt for maximizing inference throughput, and without breaking compatibility with other inference methods**. We provide a brief comparison for each work referenced in the review to clarify the most important differences:
> 1. **SpecInfer**: This method is specifically designed for speculative decoding verification. Their method does not parallelize the generation of different answers with shared contexts. Although a similar causal mask is used, their implementation (a) limits the number of tokens that can be decoded, and (b) cannot have autoregressively generated tokens appended every iteration, as they instead generate a new tree for every verification. Finally, the authors claim that SpecInfer is designed and most beneficial for low-latency LLM inference, which is not interchangeable with high throughput. IPPD is targeted for offline settings where latency is a non factor, and our evaluation is centered around using both a high batch size and high number of stacked questions per prompt to maximize throughput. The relative saturation of memory and operations and floating point operations in this regime is not comparable to the one tested in SpecInfer.
> 2. **DeFT**: This method is also specifically designed for minimizing latency, which does not align with the problem setting of IPPD. Furthermore, the implementation of DeFT, although very efficient, requires a complete overhaul of the attention module and manipulation of the KV cache, which limits compatibility. The method only provides a Llama implementation, and does not currently support batching multiple independent trees, which is important for high throughput. IPPD is compatible with any LLM capable of receiving an attention mask and position IDs as input parameters, which is standard practice in the Hugging Face ecosystem. As no code needs to be modified in the model architecture or attention kernel, IPPD does not interfere with any existing attention kernels, KV cache or batching functionality. Lastly, DeFT disregards the use of dense causal masks because of the additional memory accesses required to load them. Our results demonstrate that such masks can be very efficient, as it completely removes the need for secondary reduction operations on partial attention results, and the memory access of loading the mask is far outweighed by the significantly larger KV cache in real world applications.
> 3. **FastTree**: FastTree does not combine multiple questions into one fused kernel with a causal mask like IPPD. It strongly resembles Cascade Inference using partial attention computation and reduction operations, which we discuss in Section A.2.3. It is additionally limited by only accelerating autoregressive decoding after prefill, which would not provide any speedup for two of our benchmarks featuring single-token generation. Despite these important differences, we acknowledge that a direct comparison with this method would be beneficial to further demonstrate the effectiveness of IPPD against a strong baseline. We will work to include these results in a revision of this paper.
> 4. **FlashForge**: IPPD treats the KV cache as regular 4D tensors, by combining questions with common contexts into a single cache entry. This does not result in increased memory usage, and makes the cache reads contiguous. This is a fundamentally different approach than storing the KV cache using a tree structure of irregular 3D tensors, like in FlashForge. Furthermore, comparison with FlashForge is infeasible without publicly available source code.
>
> Regarding your question about the vLLM baseline: The two main features of vLLM we are comparing against IPPD is prefix caching and paged attention, neither of which require the asynchronous decoding engine of vLLM. The synchronous vLLM engine is able to cache the shared prefix of concurrent requests, and efficiently store/load the KV cache without creating duplicate entries. The asynchronous mode offers completely orthogonal speedup by using continuous batching, a feature that is compatible with IPPD but not available in the Hugging Face framework. This integration is a direction we are interested in pursuing in future work.
>
> We hope that we have addressed all of your concerns about related work and experimental details, and are happy to engage in further discussion to improve your confidence in our work. We plan to follow-up with a revised version of the paper including the expanded literature review and experiments with FastTree/Cascade inference prior to the end of the discussion period.

---

> > ### Comment · Reviewer_1iVT · 2025-11-24
> >
> > > "The novelty of IPPD is providing an efficient method for stacking questions with common context into a single prompt for maximizing inference throughput, and without breaking compatibility with other inference methods."
> >
> > I am still a bit confused. What is the main difference of this kind of stacking with tree-based decoding? According to my understanding, the IPPD is just a special case of tree-based decoding, as discussed in these prior works designing efficeint kernels for tree-based decoding.

---

> > > ### Author Response · Authors · 2025-12-03
> > >
> > > Thank you for the follow-up. We would like to clarify the relationship between IPPD and the concept of tree-based decoding with regards to novelty.
> > >
> > > *We view tree-based decoding as a broad **problem category**, of which the problem setting for this work is a specific case*. We are given a set of token trees, where each node contains an ordered sequence of tokens, and edges between parent and leaf nodes indicate a causal dependence. Tree-based decoding requires computing self-attention scores on token trees, imposing the following criteria: (1) tokens within a node must attend to all preceding tokens in that node's sequence, (2) tokens within a node must attend to all tokens in ancestor nodes, and (3) the token tree must be expandable to include newly sampled tokens at every decoding step. Given the widespread appearance of tree structures in practical LLM generation scenarios (eg. CCQA), it is natural that there exists several works presenting distinct methodologies to solve the same problem as efficiently as possible, each with their strengths and constraints.
> > >
> > > IPPD is a method that solves the tree-based decoding problem with (a) the optimization objective of maximizing throughput and (b) the constraint of maintaining compatibility with as many other inference acceleration methods as possible. The IPPD method itself does not involve the use of any tree structures: From the memory view, the tokens and corresponding KV cache are stored in contiguous tensors without any hierarchy. From the computation view, self-attention proceeds as usual with a single dense matrix-matrix multiplication between queries and keys. The structural information of the original token tree is completely encoded within the custom attention mask and position IDs, thus allowing the tree to be completely flattened and treated like any regular token sequence. This is in contrast to methods such as DEFT or FastTree which maintain a tree structure in both the memory and computation view, requiring a custom attention kernel. IPPD is specifically designed to provide a large throughput increase to existing models whilst being agnostic to the  already optimized implementation of `model.forward(input_ids, position_ids, attention_mask, kv_cache)`. We therefore argue that the solution provided by IPPD for the tree-based decoding problem is (1) distinct in its optimization objective and constraints, (2) novel in its tree-free implementation, and (3) highly applicable to existing production systems without custom model or kernel integration.

---

### Official Review · Reviewer_uN7x · 2025-10-31

**Soundness:** 3
**Presentation:** 1
**Contribution:** 3
**Rating:** 6
**Confidence:** 3

**Summary:**

Intra Prompt Parallel Decoding (IPPD) proposes packed matrix multiplication computation to speed up jagged prefill and decode requests.

**Strengths:**

I think IPPD is quite efficient when the questions and decoding parts are brief, because we do not need to implement a jagged tensor, which often results in lower utilization of TensorCores (matrix multiplication accelerators).

The method is very intuitive and simple.

**Weaknesses:**

I think this method is only effective when the question (later part prompt) and the answer are extremely short (<128).
However, such a scenario is extremely rare in the agentic AI era. Therefore the effective-ness of this method is pretty limited.

**Questions:**

### Questions
- What if the decoding length is long? e.g., reasoning models
- What if prefill is always large? e.g., tool calling

### Formattings
Can you update the figures to make them more intuitive about your method?

---

> ### Author Response · Authors · 2025-11-21
>
> Thank you for the positive review and for highlighting the efficiency of our method within the problem setting. You are correct to point out that although IPPD can be applied to any length of prompt, it is most effective when the ratio between the length of the common context and the question/answer pairs is high. This high ratio is actually quite common in applications that share context for multiple questions/queries:
> 1. Extractive QA tasks usually require short answers, as we are looking to extract a single piece of information from a large body of contextual information, for example a book or manual.
> 2. Agentic AI systems often contain a large amount of common context in their system prompts and tool calling function definitions.
> 3. Although reasoning models have been shown to increase performance, there is a tradeoff of increased computational cost. This has led to significant research on how to shorten or parallelize thinking chains in reasoning models (Qu et al., 2025), which can better utilize IPPD. Other applications such as Attribute Value Extraction in the shopping or web search domains can contain millions of documents and queries (Hongwimol et al., 2025), making short answers a necessity due to cost constraints.
>
> Our experiments show that existing baselines tend to underperform in these conditions, so IPPD provides a clear improvement in this important setting.
>
> Regarding your second question, long prefill can become the throughput bottleneck when single token answers are required. For answers requiring multiple tokens, autoregressive decoding remains as the key bottleneck. Methods such as chunked prefill can help mitigate the prefill cost, and are completely orthogonal to IPPD.
>
> We are also committed to improving the clarity of our work through more intuitive figures. Would you be kindly able to provide additional feedback on how they could be improved?
>
> We hope that our response addresses all of your concerns about the applicability of our method in real-world scenarios, and are happy to engage in further discussions to aid in improving our work.
>
> **References:**
>
> Pollawat Hongwimol, Dong Sheng, Li Zhang, Kai Liu, and Xiufei Wang. GAVEL: Generative attribute-value extraction using LLMs on LLM-augmented datasets. In *Proceedings of the 4th International Workshop on Knowledge-Augmented Methods for Natural Language Processing*, pp. 81–90, May 2025.
>
> Xiaoye Qu, Yafu Li, Zhaochen Su, Weigao Sun, Jianhao Yan, Dongrui Liu, Ganqu Cui, Daizong Liu, Shuxian Liang, Junxian He, Peng Li, Wei Wei, Jing Shao, Chaochao Lu, Yue Zhang, Xian-Sheng Hua, Bowen Zhou, and Yu Cheng. A survey of efficient reasoning for large reasoning models: Language, multimodality, and beyond. arXiv preprint: arXiv 2503.21614, 2025.

---

### Author Response · Authors · 2025-12-03
**Combined Response to Reviewer 1iVT and Reviewer PotS**

We would like to address the comments from Reviewers 1iVT and PotS simultaneously regarding the inclusion of Cascade Inference and similar tree-based attention baselines:
While we agree that such comparisons could be valuable, we found that executing these methods in our evaluation setting was infeasible despite extensive effort. Importantly, the obstacles preventing comparison stem from the baselines’ own model-specific and system-specific constraints, rather than limitations of our evaluation.
- **Cascade Attention (FlashInfer)**: Only the low-level kernel is provided, with no integration for modern LLMs. The referenced “MLC-Serve" implementation from their blog post does not contain references to Cascade Attention in the documentation or code, and community questions on this remain unanswered. As a result, there is no runnable implementation available.
- **DeFT**: The implementation is tightly coupled to Llama and cannot be used with any of the models we are able to evaluate. Porting DeFT to other models would require substantial re-engineering and is not supported by the released implementation.
- **FastTree**: The implementation of the method depends on strict architectural requirements. For example, the Q/KV head counts of the model must be powers of two and greater than 16, which none of the Qwen 3, OLMo 2, Phi 4 or GPT-OSS models satisfy. The practical applicability of the method is further limited by its dependence on an outdated version of SGlang that does not build reliably under current toolchains.
- **FlashForge**: No public implementation is available.

In summary, these baselines are not general-purpose methods but highly model- and system-specific. In contrast, IPPD does not suffer from these limitations, as the method is model-agnostic, easy to integrate, and works without specialized kernels. The baselines used in our paper therefore more accurately represent the practical alternatives available to users and researchers.

---

### Author Response · Authors · 2025-12-03
**Summary for the Area Chair**

We thank the reviewers for their thoughtful feedback and would like to summarize how we addressed the main concerns regarding practical applicability, novelty, and experimental completeness.

**Practical Applicability**: We clarified that IPPD specifically targets offline, high-throughput scenarios where structured inputs are standard and cost-efficiency drives the need for short answers. This setting is common in modern applications such as document analysis, RL training and agentic systems.

**Novelty**: We clarified that IPPD presents a novel and distinct solution to the tree-based decoding problem, centered on maximizing offline throughput. This is a crucial objective that differs significantly from the latency minimization goals of prior work (SpecInfer, DeFT). The core novelty lies in its tree-free implementation: we flatten the token tree into contiguous tensors, encoding all structural information solely within the custom attention mask and position IDs. This design makes IPPD model- and kernel-agnostic, delivering high throughput while ensuring compatibility with existing LLMs and highly optimized attention routines, unlike custom-kernel-dependent methods.

**Experimentation**: We justified our baseline selection by detailing the specific architectural constraints that make methods like DeFT and FastTree infeasible for a fair comparison. Consequently, our evaluation compares against the most viable, general-purpose alternatives currently available to practitioners.

We hope this summary assists in your final decision.

---

### Meta-Review · Area_Chair_QFca · 2026-01-08

**Summary:**

This paper proposes Intra-Prompt Parallel Decoding (IPPD), a simple inference technique that enables LLMs to answer multiple questions sharing the same context within a single prompt, achieving significant throughput gains without model or kernel modifications . Reviewers acknowledge the method’s soundness, simplicity, and strong empirical speedups on multiple QA benchmarks. However, concerns are raised about limited practical applicability, reliance on structured offline settings, and, most importantly, novelty, with several reviewers arguing IPPD is a special case of prior tree-based decoding methods and lacks direct comparison to strong baselines. The authors provide detailed clarifications and justify baseline choices, but some novelty concerns remain unresolved. Overall recommendation: Reject.

**Reviewer Concerns:**

**Reviewer Concerns Assessment**

Addressed concerns:
(1) Practical applicability: The rebuttal clearly positions IPPD in offline, high-throughput scenarios and provides concrete examples (document analysis, agentic systems, RL), addressing concerns about unrealistic use cases.
(2) Baseline feasibility: The authors convincingly explain why Cascade/DeFT/FastTree are difficult to include due to lack of public or general-purpose implementations.

Outstanding concerns:
(1) Novelty: Despite clarifications, it remains unclear whether IPPD is fundamentally distinct from existing tree-based decoding methods or mainly a special case with different engineering trade-offs.
(2) Experimental completeness: The absence of direct comparisons with strong tree-based baselines (e.g., Cascade or FastTree) is still a weakness acknowledged but not resolved.

**Reviewer Scores:**

Reviewer uN7x: Likely unchanged or slightly more positive. The rebuttal addressed applicability concerns and clarified effective settings, but limitations for long decoding remain.

Reviewer 1iVT: Unlikely to change. Core novelty concerns persist despite detailed explanations, and the reviewer explicitly remained unconvinced in the follow-up.

Reviewer PotS: Possibly a small increase but still negative. Clarifications on offline use and structured inputs help, but novelty and missing baseline comparisons remain unresolved.

---

### Decision · Program_Chairs · 2026-01-26

Reject